# Our Healthy Clarence: A Community-Driven Wellbeing Initiative

**DOI:** 10.3390/ijerph16193691

**Published:** 2019-09-30

**Authors:** Nicholas Powell, Hazel Dalton, David Perkins, Robyn Considine, Sue Hughes, Samantha Osborne, Richard Buss

**Affiliations:** 1Centre for Rural and Remote Mental Health, University of Newcastle, Orange 2800, Australia; hazel.dalton@newcastle.edu.au (H.D.); david.perkins@newcastle.edu.au (D.P.); samantha.osborne@health.nsw.gov.au (S.O.); 2School of Medicine and Public Health, University of Newcastle, Callaghan 2308, Australia; robyn.considine@newcastle.edu.au; 3New School of Arts Neighbourhood Centre, South Grafton 2460, Australia; ourhealthyclarence@nsoa.org.au; 4Mental Health & Drug and Alcohol Services, Northern NSW Local Health District, Lismore 2480, Australia; richard.buss@ncahs.health.nsw.gov.au

**Keywords:** wellbeing, community-driven initiative, mental health capacity building, collaboration, public health, community development, mental health promotion, suicide prevention, rural

## Abstract

In 2015–2016, the Clarence Valley in Northern New South Wales, Australia, experienced an unexpectedly high number of deaths by suicide, and the resulting distress was exacerbated by unhelpful press coverage. The local response was to adopt a community-wide positive mental health and wellbeing initiative. This paper describes the process and achievements of the initiative called ‘Our Healthy Clarence’. Key stakeholders were interviewed at year two and relevant documents reviewed. Data were analysed using document and thematic analysis. Our Healthy Clarence was established following community consultation, including forums, interviews, surveys and workshops. It adopted a strengths-based approach to suicide prevention, encompassing positive health promotion, primary and secondary prevention activities, advocacy, and cross-sectoral collaboration. A stakeholder group formed to develop and enact a community mental health and wellbeing plan. Factors contributing to its successful implementation included a collective commitment to mental health and wellbeing, clarity of purpose, leadership support from key local partners, a paid independent coordinator, and inclusive and transparent governance. Stakeholders reported increased community agency, collaboration, optimism and willingness to discuss mental health, suicide and help-seeking. Our Healthy Clarence draws ideas from mental health care, community development and public health. This initiative could serve as a model for other communities to address suicide, self-harm and improve wellbeing on a whole-of-community scale.

## 1. Introduction

Rural communities in Australia are diverse, with strengths and weaknesses that influence mental health, wellbeing and suicide. Rural residents have been reported to have higher social capital, community identity and life satisfaction [1,2,3,4]. These strengths can be harnessed to address the challenges of rural living, one of which is the comparatively lower access to and use of health and social services [5,6,7]. This is due to there being fewer service providers per capita in rural areas, and increased distance and opportunity costs for service users. Sociocultural factors, such as norms of self-reliance, are stereotypically stronger in rural and remote areas and may also contribute to lower service use [8]. These factors, combined with overall rural socioeconomic disadvantage, contribute to the disparity between rural and urban health outcomes, such as reduced life expectancy and higher rates of disease, injury and suicide [7,9,10,11,12,13].

Despite the similar prevalence of mental illness across Australia, suicide rates are 50% higher in rural and remote populations compared to the capital cities [14,15,16]. This suggests that different strategies are needed to prevent rural suicide. The Centre for Rural and Remote Mental Health identified five areas for attention in its position paper on preventing rural suicide [14]. These include frontline responses (clinical intervention and postvention support), with longer-term primary prevention (both universal and selective), building protective factors in children and young people, and the promotion of community wellbeing including approaches that address the broader social determinants of mental health [17,18]. The latter is important, as many risk factors associated with poor mental health and wellbeing are community-level factors, such as social isolation and a lack of opportunities [19]. In addition to the 3128 people who died by suicide in 2017 in Australia, there were many more who attempted suicide, self-harmed, had suicidal thoughts or suffered from debilitating poor mental health [20,21]. A comprehensive approach which encompasses the promotion of mental health and wellbeing, primary and secondary prevention, clinical intervention and postvention will help those who are suicidal, those at risk and may prevent others from becoming so [17,18].

Prevention and promotion initiatives aim to change individual and community behaviour across settings. Principles from public health and community development can help structure these approaches by focusing on behaviours and their associated factors, the social determinants of health, and policies and practices of organisations and community groups [19]. The promotion of mental health in a rural community is a task that is too large and complex for a single organisation or profession [22]. Public health or community development initiatives are often classed as bottom-up (community-led) or top-down (government-led). The advantage of bottom-up initiatives is community appeal, as the initiative is driven by their priorities, which may promote agency [23,24]. However, these groups may lack authority or the organisational capacity to alter the service landscape. For action to occur on the social determinants of mental health, there must be both community buy-in and the ability to effect social and structural change. A collaborative group that includes the range of perspectives, experiences and capacities mentioned above should be well-placed to implement a more comprehensive approach to community wellbeing by focusing on behaviours, determinants and structural and policy changes [25]. 

This study was undertaken to examine the implementation of a community-driven mental health and wellbeing initiative in Northern New South Wales, which began in response to a geographic cluster of local suicides. The two aims to this project were to: (1) describe the initiative, its context, process and factors; and (2) analyse community and stakeholder perceptions of implementation.

## 2. Methods

The formative evaluation, namely, “a rigorous assessment process designed to identify potential and actual influences on the progress and effectiveness of implementation efforts” [26], adopted a mixed-methods approach. The purpose was to track and describe the implementation of a mental health and wellbeing initiative, and to identify factors associated with implementation and perceptions about the impact on the community. The authors recognise that there are multiple definitions of community, for the purpose of this study, community means a group of people bound by a geographical area, in this case the Clarence Valley. The purpose of this formative evaluation was not to demonstrate the success of the initiative as it is premature to make such claims. However, this initiative was of interest as it is a novel, low-cost, small, bottom-up, locally-driven approach which has demonstrated a level of sustainability and opportunistic advocacy. The perceptions from participants about the success of the initiative may relate to their internal justification to continue their activities. This is an observational study. Due to the situation within the community in 2016, it was not ethical to create an experimental trial which would have disempowered the community and delayed the response, or to gather baseline data in a community that was in crisis. Temporal comparisons are necessarily based on (a) publicly available data (which is limited), (b) documentation (which is limited), and (c) subjective reflections by stakeholders.

Data collection included a review of 65 project documents, which were the primary data source for aim 1 and quantitative information (training numbers etc.), and completion of 36 semi-structured interviews, which were primarily used for aim 2 and establishing context. The initiative’s Project Coordinator and steering committee members identified key informants and shared relevant documentation. The purposive sample included stakeholders with direct (steering committee members) and indirect (lay community members, volunteers and service providers) involvement with the initiative including representatives from the community, education, police, primary and secondary health care services, local government and various local services with an interest in mental health. Participants were also given the opportunity to nominate other stakeholders for an interview, so that recruitment could snowball. Of those interviewed, 64% (23/36) had steering committee experience.

Interviews were recorded, transcribed, de-identified and stored using NVivo 11 [27]. The quotes that are in this manuscript were selected as they represented a theme from the qualitative analysis. Document and inductive thematic analysis were used to identify themes and concepts within the transcribed interview data and the documentation [28,29,30]. This enabled the retrospective generation and validation of a program logic model and a key events timeline [31,32]. Feedback, verification and validation of findings were sought from key team members in a face-to face workshop and during successive iterations of this manuscript (RB, SH and SO). The project was approved by the University of Newcastle human research ethics committee (H-2017-0421).

## 3. Results

### 3.1. Clarence Valley Context

The Clarence Valley Local Government Area (LGA) has 51,570 residents and covers 10,441 km^2^ on the northern coast of NSW in Australia (see Figure 1). The LGA is served by the Northern NSW Local Health District (LHD) who deliver secondary and tertiary care, with approximately 300,000 residents across 20,732 km^2^ and the North Coast Primary Health Network (PHN), who support primary care, with approximately 520,000 residents across 32,047 km^2^ [33,34].

In 2016, the Clarence Valley scored 926 on the Socioeconomic Indexes for Areas (SEIFA) for disadvantage, which places it in the lowest quintile in NSW [35]. Between 2015 and 2017 there were 97 hospitalisations for self-harm in the Clarence Valley (187.8 per 100,000) which was appreciably higher than the NSW rate (104.5 per 100,000) [36]. This represented an appreciable increase over the previous 5 years. In 2016, the Northern NSW LHD, as a whole, had an appreciably higher suicide rate than the rest of NSW (17.8 and 10.0 per 100,000, respectively) [36]. It is difficult to determine the exact number of deaths by suicide in the Clarence Valley by year, but it was perceived by the community and media that there was a geographical cluster, and this represented a contagion.

### 3.2. Our Healthy Clarence

In early 2016, the LHD was lobbied by the community and services to take action. In response, the LHD and PHN organised an interagency meeting to consider the next steps for the Clarence Valley community. This meeting consisted of primary care, mental health, social and voluntary services from the Clarence Valley region who were members of other mental-health-related interagency groups, as well as external consultants. As a result of these initial meetings, the perceptions of the broader community were sought. Semi-structured interviews were conducted with 99 local stakeholders by a Centre for Rural and Remote Mental Health associate, who was external to the community and was seen as impartial. The interviews revealed that the community were committed to addressing mental health issues within the community, but that there was a low awareness of available services and poor communication between services. A series of community workshops discussed the findings from the interviews which began to orient the community towards action, rather than problem description. Consultants also presented local health, social and service data, coupled with several evidence-informed models and frameworks on public health, mental health and wellbeing, and suicide prevention (including Lifespan) [37,38,39,40]. These workshops were attended by over 100 community members who had a broad range of backgrounds and experiences. The initial vision for the community was a focus on mental health and wellbeing rather than a narrow suicide prevention committee. A steering committee was formed where members were nominated by the community, members needed to have the time available to commit to the group. The steering committee developed a plan for improving mental health and wellbeing in the Clarence Valley, based on the workshops, iterative feedback and a vote.

The 2016–2018 mental health and wellbeing plan [41] had five objectives:To improve access for people at risk of self-harm to treatment, crisis care and care after an attempt.To improve the ways in which workers and the community respond to people at risk of self-harm.To ensure that suitable mental health and wellbeing programs are available in schools.To improve community awareness of mental health. This includes how to access information and services.To improve our connection with the community. To improve early support for people who are at risk of self-harm and to help prevent self-harm.

The initiative was named Our Healthy Clarence (OHC) by the participants of the community workshops. The OHC steering committee includes community members, service providers and government department representatives. Once the plan was agreed and published, working groups were created to implement each of the strategies. These groups included steering committee members as well as other key stakeholders (e.g., School principals for objective 3). Community ownership was reinforced by the community endorsement of the initiative and its logo, which were voted on by the community.

### 3.3. Description and Progress of Our Healthy Clarence

The early success of OHC included a collective approach to advocacy and improving access to help and care, yielding numerous new services including a *headspace* centre and pop-up information and referral hubs in community centres.

A retrospective program logic model of OHC was constructed from the documentation analysis, including the original OHC plan, coupled with additional content from the stakeholder interviews. The activities mapped consistently to the five objectives in the 2016–2018 plan [41] (see Figure 2). The purpose of the figure is to illustrate the breadth of activities undertaken and capture their strategic alignment with the OHC plan objectives in order to understand the intervention and its components.

Analysis of documentation and interviews have informed a timeline of key events and phases that have shaped the initiative. Figure 3 shows how OHC developed over time. The inception phase included a range of engagement activities such as community forums, interviews, a community report and a planning workshop with a mix of subjective and objective data. The agreed plan implemented over time included key events such as the funding of services identified in the original community engagement report and the appointment of an OHC Project Coordinator funded from the pooled resources of several participating organisations.

Funding to address service gaps was attained through advocacy and proposals by OHC and the broader Clarence Valley community. Funds were allocated to support the establishment of new services in the region, including a *headspace* clinic in Grafton and an aftercare service, not to directly support the activities of the steering committee. 

### 3.4. Summary of Key Contributory Processes and Identified in Documents and Interviews

The following table provides a short summary of the themes and sub-themes identified in the analysis of documents and interviews (Table 1). Each theme is presented in more detail below. It should be noted that many of the themes are an articulated reflection of overcoming obstacles or the identification of the absence of an enabler.

#### 3.4.1. The Community-Owned, Codesigned Approach Promoted Engagement and Empowerment

From its inception, community governance was central to OHC. This reflects a strong bottom-up approach to community development, where the community decides what they want to achieve. This approach stimulates “buy-in” for the initiative and can be a powerful impetus to get the initiative started.


*‘… there was a sense that, for it to be successful, it had to come from the community themselves and not someone coming in from a couple of hundred kilometres away, or two hours away, saying how to do things’—Participant 26*


The early consultation meetings, workshops and interviews provided pathways for community input. This approach enabled community feedback through multiple channels and empowered key community members to represent sectors of the community and stay in tune with issues as they changed. This cross-cutting advocacy and communication worked across all areas (see Figure 2).

One of the advantages of this bottom-up impetus is that it builds on community interest and drive. In this case, the community were motivated to do something about the suicides in their community and to examine broader wellbeing. As such, the initial engagement by lay community members in the planning was high. The early energy was harnessed to create the community plan (via community interviews and planning workshops). Participants recognised, however, that this form of engagement dissipated quickly.


*‘I think what hasn’t worked well is … keeping the community engaged. Because community were very engaged to start with, extremely engaged. They are the ones that drove that meeting or those three meetings where over 100 people came to each one. More and more, as experts came in, those community members petered out and just went, okay, so we now we know the experts are involved, as does happen, so the engagement didn’t continue.’—Participant 1*


The engagement in activities and training has remained strong, but direct lay community member involvement in the steering of OHC was lower than initially planned. Six places were allocated for lay community representatives, with only two taken up, and a further three youth representatives joining more than a year later. While this was noted as a weakness by some, other participants underscored that the majority of steering group members were local residents, irrespective of the services they represented. The majority of the participants wanted to see more community representative involvement in the steering committee, but realised it was difficult to bring new people into the initiative, after the community urgency had dissipated.


*‘It actually needs to have a good cross-section of community engaged people sitting on steering, who are from community and not from the services... So I think… the steering membership should have a stronger community presence.’—Participant 18*



*‘I would say the opportunity missed, in my view, is harnessing the community passion in an effective way early on.’—Participant 6*


Community engagement fluctuated over time, reflecting variable community readiness and perceptions of sustainability. For a community to take a bottom-up approach, there must be interest and motivation in the subject. Often this arises through adversity, such as natural disasters or social disadvantage. The issue of whether a crisis is necessary to stimulate an initiative like OHC divided participants. Some believed that community drive should be harnessed as it arises, while some thought it could be manufactured through the advocacy of a passionate community group.

#### 3.4.2. The Initiative Took a Strengths-Based Approach to Suicide Prevention Via Wellbeing

OHC took a community wellbeing and strengths-based approach to their planning which aimed to build mental health, mental health awareness, capacity to recognise and respond to declining mental health and reduction of stigma across the community. The committee was determined to maintain its priorities and ownership, as external input increased. Throughout the initiative, the committee has been pressured to take an individually focused, narrow approach to suicide prevention. Participants recognised that the wellbeing approach was helpful for themselves and for the engagement of community members, which was clear in the strategic objectives (see Figure 2).


*‘I think it’s about taking the focus away from suicide and bringing that into resilience-building and fostering hope within a community and done through participation, different activities, activities available to young people, things that they can hook into and find meaning from.’—Participant 30*


This approach allowed the steering committee to work towards mental health, wellbeing and reducing suicide, without feeling responsible if and when a suicide did occur and being prepared for such an event. This approach made the initiative relevant to all community members, as everyone is able to take steps to improve their wellbeing. This shift made mental health and wellbeing part of the conversation throughout the community and gave them a broader scope to make use of opportunities such as mental health month. The approach helped to empower the steering committee as it allowed them to see what was good in their community, rather than focus solely on weaknesses.


*‘The overall effect has been to move … from a deficit kind of conversation to a strength-based hope-filled conversation.’—Participant 35*


The other aspect of the strengths-based approach focused on community assets. Many communities have underutilised services which are ineffective or disjointed. The early stages of OHC greatly improved the awareness of services and service collaboration, which increased their ability to obtain funds for programs and decreased duplication.


*‘You never know what you’ve got until you actually look about what you have, as far as services go. I mean a lot of areas may or may not have a range of services but they will have some. So, find your strengths; find those local services that know your area, are made up of workers from your area, to really give that focus.’—Participant 2*


#### 3.4.3. Governance and Structure were Important to the Success of the Initiative and Matured over Time

Participants recognised that the people involved in OHC were crucial to its success. However, it was the way these people applied their skills, experience, resources and networks that made the initiative viable. Firstly, the Local Health District hosted the community forums despite reputational risks. This gave the community a chance to have their voices heard by decision-makers.


*‘I have been here a long time; I am completely embarrassed because I am going to a meeting I have no answers for. And then I think well, how would a community member feel?… But somebody has got to lead it. You can’t wait…’—Participant 27*


Secondly, one of the early barriers to bottom-up, community initiatives is the loss of direction due to limited structure, documentation and professional systems. Initially, the LHD and PHN supported the formation of the Mental Health and Wellbeing Plan and provided professional and secretarial skills for the steering committee, which instilled processes of professionalism that have helped the steering committee to function (see Figure 2). This included a formal application process and terms of reference for committee, membership which provided legitimacy and helped to manage membership, conflict and action. The professional involvement also helped to lead the initiative initially and build leadership qualities in the steering committee.


*‘My first response was of frustration, because I could see that around the table, we didn’t have the solution, because we didn’t have the decision-makers. But very quickly from there, the initiative grew into something that was actually actively led and supported by people who were more senior, in particular, government organisations.’—Participant 13*


Thirdly, there was the function of the steering committee itself. The steering committee brought Clarence Valley Council, LHD, PHN, services, government departments and community members together to examine all aspects of wellbeing promotion. It comprised of 15–20 members throughout, and although the members have changed, the core philosophy remained. The group were responsible for developing both iterations of the Mental Health and Wellbeing Plan, which have been key tools in articulating the vision to partners, community and government. The steering committee now has a track record of planning, action and results. Participants reported that it was able to effect change and respond to adversity as it arose. As the steering committee matured, it has targeted particular groups and organisations who can contribute to the vision but has maintained control over the processes that occur under the name of OHC. Although the process was not quick or easy, it enabled decision making and local control over vision and direction.


*‘The early days of the steering committee was very labour-intensive, and it was, in the most part, an adjunct to our day jobs. So, a lot of energy went into it and at one point there was a core group of local service providers that had dubbed it Our Unhealthy Clarence, because we were almost obsessed with it. But it’s kind of that kind of energy, I think that’s needed to get things off the ground.’—Participant 35*


The final sub-theme was the structure adopted by OHC. Each objective had a working group with steering group members and the broader community that reported to the steering committee (see Figure 2). These working groups gave people a broad remit for action while maintaining accountability. It allowed the flexibility to ‘go where they were needed’. For example, one member visited schools, media, community events and services to garner community support in the most acceptable way. The coordinator position proved important in facilitating community engagement.


*‘I feel that what’s worked well is having a coordinator; that’s made a huge difference. I think a lot of the services are already quite strained … It’s helped in my old role to have the flexibility that I had to be able to respond to social issues.’—Participant 4*


By early 2018, it was clear that the working groups had become less effective as membership declined. This was attributed to the burden participating along with unclear guidelines for action. In the next iteration of the plan, the working groups are being redesigned as time-limited implementation teams, giving volunteers clearer objectives and time-limited involvement.

The question of steering membership divided opinion. Some participants felt that it would become ineffective if it became too large, but others felt it important to include more community members. There was an interesting tension between inclusivity and effective committee size. Although the steering committee membership has not increased significantly since 2016, the membership profile has altered. The LHD and PHN now take a smaller role and some senior members have reduced their participation. However, over this time there has been an increase in community representation. This may represent a changed focus for the initiative, as it becomes a more community-driven. Key decision-makers were important in the formation of the initiative and in establishing the structure, but as the committee moved to the promotion of wellbeing, perhaps the voices of the community needed to become stronger. One participant also raised the issue of confidentiality on the steering committee. It was important to balance confidentiality with transparency. The OHC has evolved to better suit the needs of the community, and so the membership has changed accordingly.

#### 3.4.4. The Culture of Collaboration Increased Trust, Coordination and Agency

Collaboration between services and other members of OHC was seen as a major reason behind the improvement of services, care and community activities. While it was difficult for the community to establish the initiative, greater service collaboration was achieved and created a platform for change.


*‘I think the services themselves though provide a good starting point to enable action. I think any community is unaware of where to start, it’s quite overwhelming. I think that there needs to be initially that service collaboration to discuss where to from here, in the same way that we did.’—Participant 4*


This collaboration included services sharing information, advocating for and collaborating on bids for new services, and arranging community events. Nonetheless, this collaboration made the services better at contributing in ways that were good for the community. Personnel and structural changes in collaborating organisations have been challenges for OHC, and at times caused a decrease in impetus and cohesion.


*‘There’s been some leadership changes and unfortunately the people that were involved with this mustn’t have kept terribly good notes because [some organisations are now] largely unaware of what Our Heathy Clarence is about.’—Participant 16*


As the committee learned to collaborate, they became a powerful and assertive entity for liaising with external agencies (such as government, private organisations who deliver mental health services by contract, and non-governmental mental health organisations), so that more activities were organised with the voice of the community understood and listened to, rather than being driven by the priorities of the external agents. The example of schools working together was powerful. The principals of all Clarence Valley High Schools, now meet regularly and have a consistent strategic plan for mental health and wellbeing. This has meant that children who move between schools receive consistent and continuous care, if needed. It also means that the principals had established protocols to follow and a network of support.

#### 3.4.5. The Activities of the Initiative Consistently Reflected the Community Vision

Due to the strengths-based focus of OHC, it was important to participants that the initiative promoted positive wellbeing through its activities, which should be community-oriented (see Figure 2 for overview). However, the participants also recognised that the way services were delivered needed to change to improve access to care. New services such as *headspace* and the *Way Back Support Service* were seen as key community and steering committee achievements and successful advocacy. The entry into services was also made easier and less daunting.


*‘I wouldn’t know about half of the services if Our Healthy Clarence hadn’t happened. And that’s coming from someone who already was studying to work in the sector.’—Participant 2*


As the initiative has matured, it has become more community-oriented, and the addition of a full-time coordinator was seen as key in growing the initiative in the community.


*‘The steering committee decided that no service provider at that time or community member really had the capacity to lead that plan, so it was decided that there was a need for a paid worker, and so five organisations committed to co-funding that [coordinator] position and that just made a significant difference…’—Participant 4*


As the plan was implemented, more activities and safe spaces were provided in the community. The creation of community pop-up hubs, where Youth Workers were employed to help people in a range of circumstances, was seen as a valuable way to engage with the community. The hubs offer a range of activities for a number of demographics but have been particularly successful in giving young people a safe place. The staff have found that they can develop relationships with the visitors which enables them to hear about the stresses experienced. They were then able to refer to relevant community services, whether for mental health other needs, such as housing or financial counselling. The two pop-in hubs had nearly 3000 ‘drop-ins’ in the first quarter of 2018.


*‘It has taken a little while just to get the young people used to the hub and used to the people that work here and to get to open up a bit, but what makes us different is we’re running with youth workers in charge as opposed to just volunteers or centre staff. So if we overhear a conversation or if a young person approaches us with concerns that they have we’re able to address them by referring them on to services.’—Participant 14*


Participants recognised that engaging the whole-of-community has been a challenge as community members have other priorities. However, there have been successful ways to engage with groups within the community, particularly youth and through social media. Alongside the hubs, there have been more community events about education and awareness building, particularly in Mental Health Month, as well as activities to promote social connectivity.


*‘Activities, so many activities– having come from the Hunter Valley there is nothing- there are so many activities here for the kids and it’s really pushed quite effectively through all forms of social media. Having arrived here and new to the town and not knowing anyone – we don’t have any family here or anything—I accessed that information quite quickly and really easily.’—Participant 10*


The Clarence Youth Action Group has become an important part of OHC and organises activities for young people in the community. Participants valued the role that they played as a voice for youth and as a role model for younger people.


*‘They act as a fantastic consultative body to—I mean, they’re part of the steering committee but if we’re thinking about doing a youth event, then we do it with them and they help us be on track and make sure that we’re hitting the objectives of what the young kids want. So that’s been a huge masterplan, the number of events that they’ve led and initiated.’—Participant 23*


The participants noted that community members now seemed more willing to talk about mental health, and often this was attributed to the training in the community. Under the banner of OHC, approximately 2000 people were given some form of mental health or suicide prevention training which helped them recognise distress, start conversations about mental health and realise the options for help in their community. Interviewees believed that these approaches had improved awareness and reduced stigma. Participants also acknowledged that, due to the wide range of activities of OHC, evaluation had been and would continue to be difficult.

#### 3.4.6. OHC Coincided with a Changed Community Narrative of Hope and Agency

Through the establishment of the OHC and its early achievements, participants perceived that the community had transitioned from one of fear to a community of hope. The process of community consultation and response in the form of OHC as a positive community wellbeing initiative rather than a focus on preventing suicides was key in contributing to the sense of hope (see Section 3.4.2, above).


*‘The community narrative has certainly changed. No longer do I hear we’re a community that’s been forgotten; I’m not hearing about we’re a community where all our young people are dying, that it’s a suicide town, that there’s nothing here for the kids, why doesn’t government people do something for our community? That’s changed, that’s changed. There’s been a real change in narrative where it’s now shifting to, what can we do to ensure the mental health and wellbeing. That’s the focus and that’s hard to measure but it’s certainly out there happening.’—Participant 23*


The community and OHC committee have become aware of their strengths and trust in their capacity to use them to address challenges as they arise.


*‘Now, with that collaborative stuff, there is that sense that we can work together, we can achieve anything. And that’s not just with mental health, that’s actually with a couple of other social issues that have come up recently, we’ve done it before so we can do it again.’—Participant 4*


Since the inception of OHC, there has been a concerted effort to improve the media coverage of mental health-related issues in the Clarence Valley, which has contributed to the positive community narrative. Through the OHC website, social media and through the committee and working party membership there were many positive stories about mental health and coverage of related issues has been informed by Mindframe guidelines and steering committee agreement. As a result, the tone of newspaper articles since the inception of OHC has changed towards positive and destigmatised language (see Appendix A for methodology and results). Through the use of the media, the community made it clear that they needed more support, especially when it came to youth mental health. The establishment of *headspace* and other initiatives in the early stages of OHC were key in demonstrating that the concerns of the community were being heard and responded to (see Figure 3 for timeline). This contrasted to previous attempts to address concerns about suicide which were perceived to fall on deaf ears.

The participants recognised that it is likely there will be other suicides in the community. While there were concerns about the impact of this the sense of hope, participants suggested that the community was well-placed to respond to this in a positive and supportive way.


*‘I think that heat has gone out, if you like, or the anxiety has gone out of the community, but it would return very quickly if there was another teenager take their own life down there again. But what I do think is much better now, is the network of people, and they would together really, really quickly and look at a community response, rather than an ad-hoc, lots of different, smaller responses …I think that would be an improvement as a result of Our Healthy Clarence.’—Participant 26*


## 4. Discussion

The Our Healthy Clarence (OHC) initiative is a comprehensive and upstream approach to suicide prevention. Its overarching goal is the promotion of health and wellbeing. It is a collaboration between the community, local government, health services, education, police, and community-managed organisations. While each organisation plays a significant role, the initiative is community-controlled. Understanding the process and factors for its implementation may support its continuation and inform other communities faced with similar challenges.

OHC was formed in response to a crisis. Since its inception, the community, as represented by the OHC governance committees, was resolute in keeping the initiative focused on community mental health and wellbeing. This resolve was tested by policy and program directives linked to suicide prevention which focus on an individual conception of suicide prevention. Despite these challenges, OHC has maintained its vision on community mental health and wellbeing and become a valued community initiative and this has coincided with an improvement in circumstances for the Clarence Valley community. Through active collaboration it has created a movement of community members with the confidence and capacity to address community issues in a concerted and pragmatic manner. The structure and collaborative nature of OHC have given members a reliable source of support and information for achieving change. It has been valuable for the community to see that there is an organisation who will take action or advocate for improvements in issues related to wellbeing and this has changed the community narrative about mental health, suicide and the quality of life in the Clarence Valley. This has been achieved by harnessing the community-driven vision and energy and enabling action through top-down authorisation and direction and the ability to leverage expertise when required.

The actions of OHC mirror the five focus areas of the Centre for Rural and Remote Mental Health’s Position Paper on Rural Suicide and its Prevention [14]. The initiative has improved care after a suicide attempt, through the *Way Back Support Service* and greater service collaboration. It has also increased access and pathways to care for the community through the improved mental health care at the emergency room, pop-up hubs and through extensive mental health training. These primary prevention techniques have been used in other community suicide prevention initiatives [42]. However, OHC has taken extra measures to prevent suicide by working to build protective factors in the community by building social connectedness and creating an awareness of wellbeing. This aligns with the third, fourth and fifth focus areas in the Rural Suicide Prevention Position Paper, which focus on longer-term prevention by building a healthy and resilient community, increasing protective factors in young people; and providing support to vulnerable sub-populations. The evidence from OHC supports the view that mental health services are only one part of the solution. Many of the issues that cause people to become suicidal lie in stressors and other social and economic determinants. Without addressing these factors, suicide prevention will be limited as the factors that were present before treatment will be unchanged afterwards.

The influence of social determinants on suicide means that suicide prevention, and more broadly, mental health promotion should be seen as public health issues. One of the functions of public health initiatives is to stimulate behaviour change that is conducive to health. There are two relevant issues that relate to best practice to create behaviour change and gauging community change in health promotion initiatives. Firstly, there is a tension between bottom-up and top-down community development. Bottom-up approaches are successful as they are in tune with the community and can build support more easily and seem to be key to empowerment and sustainability [24]. Meanwhile, top-down development contributes decision-making capacity and resources to the initiative. OHC had both bottom-up investment and top-down support as needed for the development of the initiative, which contributed to the sustainability of the initiative. The impetus came from the community and this aided the sustainability of engagement throughout. However, the initiative clearly gained momentum when key decision-makers hosted the community forums. Once the community had been engaged successfully, these senior decision-makers played an important role in setting the professional groundwork for the steering committee and the mental health and wellbeing plan. Over time, the Clarence Valley community have recognised that OHC is able to enact change. The participants recognised that the decision-makers were important in making action happen, particularly in the early stages of the initiative. Therefore, this initiative has moved through phases where different degrees of community consultation and engagement have occurred, as it did so, the initiative moved between top-down and bottom-up practices. As the steering committee and community increased in confidence, the initiative has become progressively more community-owned and represented.

Secondly, although participants recognised the value of evaluation, there was no clear agreement about how to demonstrate progress and impact. There are challenges to the evaluation of OHC as the initiative operates in a complex setting and pursues a multi-dimensional outcome, namely, wellbeing [43]. Participants recognised that a simple metric such as suicide rate or hospital admissions for self-harm would not adequately reflect the objectives of the initiative, which are much broader. Additionally, suicide remains a relatively rare phenomenon, and as such occurs only in low numbers in rural communities. Therefore, the suicide rate of a town can fluctuate greatly due only to a handful of cases. It would also be difficult to link the actions of OHC directly to changes in the suicide rate as broader social factors remain relevant to individuals. To date, the initiative has gathered a broad range of evidence of progress in the community. This practice is supported by other public health initiatives in complex settings, where a randomised controlled trial is neither practical nor ethical [44]. Kagan and Kilroy [45] suggested that including both objective and subjective measures is beneficial to gain a greater understanding of projects in the field of community development. For the Clarence Valley, this method would capture and value the voices of the community in their evaluation, which would help maintain the community-driven aspect that is important to the stakeholders of the initiative.

### Limitations

This study has several limitations, given that it is a formative evaluation there is limited assessment of impact and all such data is based on subjective perceptions by interviewed participants. Moreover, with a purposive and snowball sampling methodology to capture context, experiences and processes, the sample is both small (n = 36) and potentially biased to those who know more about the initiative and may not be impartial. There is no baseline data as the opportunity to study this initiative came after its inception.

## 5. Conclusions

OHC has taken a wide-ranging approach to the promotion of community wellbeing. In addition to primary suicide prevention techniques such as gatekeeper training, destigmatisation, after-care, and improved access to care, they have an overarching philosophy based on community wellbeing. It is too early to establish whether the positive focus on wellbeing has had an effect on the suicide rate, but this may be a poor impact measure due to the relative rarity of suicide. OHC has created a group of stakeholders that have changed the narrative about mental health and wellbeing in the Clarence Valley and have the agency and capacity to address issues as they arise. As such, OHC could serve as a model for other communities who wish to address mental health and wellbeing. This project has highlighted, however, that these initiatives must respond to the local context and build on local assets if they are to be relevant and sustainable. More rigorous evaluations of community-based initiatives such as OHC have the potential to inform knowledge about suicide prevention. Opportunities for more rigorous methodology and design could be explored. However, the tension between the needs of communities and the requirements of rigorous research design need to be considered [46].

## Figures and Tables

**Figure 1 ijerph-16-03691-f001:**
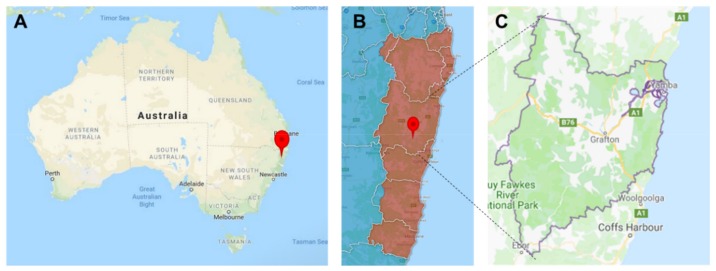
The Clarence Valley in the context of Australia (**A**, red marker); the North Coast Primary Health Network (**B**, red marker compared to the dark red shaded area) and the Clarence Valley Local Government Area alone (**C**, grey outlined area).

**Figure 2 ijerph-16-03691-f002:**
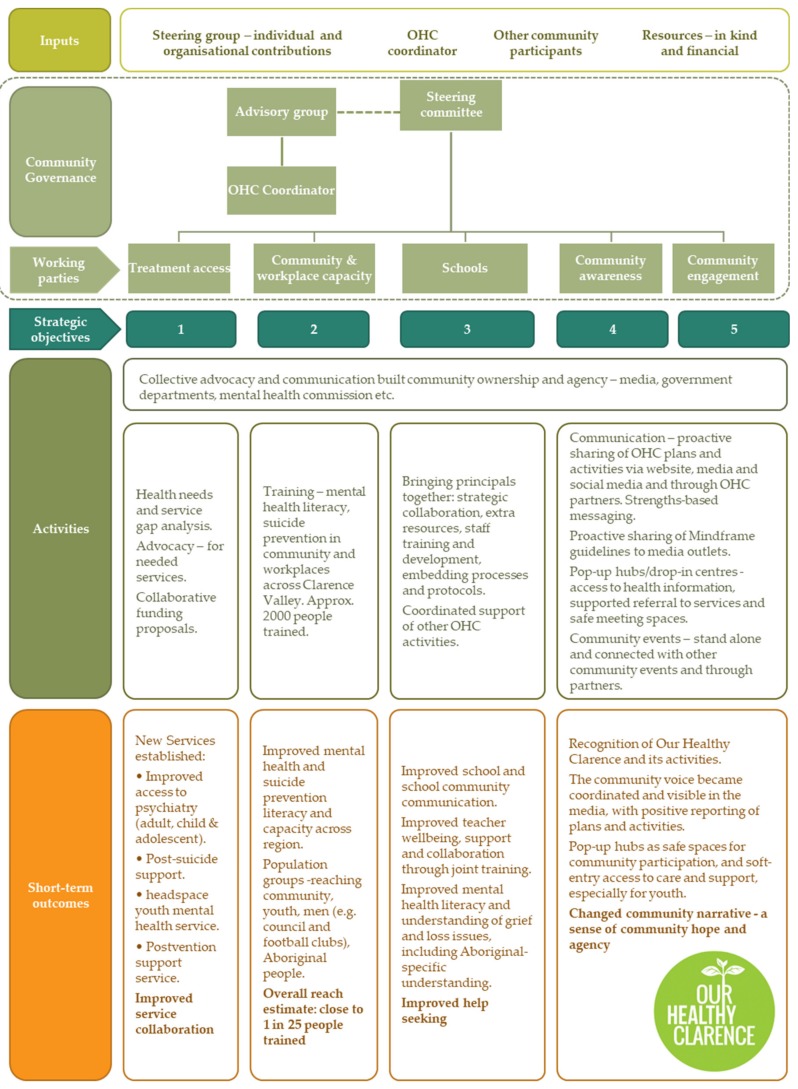
Our Healthy Clarence—what happened?

**Figure 3 ijerph-16-03691-f003:**
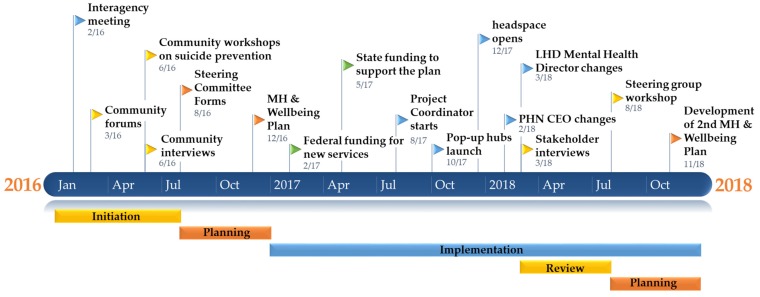
Key events and changes to Our Healthy Clarence over the planning and implementation period.

**Table 1 ijerph-16-03691-t001:** Key themes and sub-themes from the interviews and document analysis.

Theme	Sub-Theme
The community-owned, codesigned approach promoted engagement and empowerment	The community decided priorities which promoted buy-in
	Multiple channels of engagement promoted representation
	Engagement should be harnessed while it is present
	Community readiness can arise from a variety of circumstances
The initiative took a strengths-based approach to suicide prevention via wellbeing	The wellbeing approach helped the community build after tragedy
	The wellbeing approach gave a broader reach into community
	The strengths-based approach got the community to realise its assets
	Policy and programs focused on suicide prevention as a negative construct conflicted with the community desire to focus on community wellbeing as a positive approach
Governance and structure were important to the success of the initiative and matured over time	Early forums made it clear that community were drivers
	Professional support assisted the governance of the initiative
	Collaboration helped to realise the vision
	Positions were given the flexibility to adapt to community needs
	The committee had to balance inclusivity and size
	The committee had to balance transparency, sensitivity and confidentiality
The culture of collaboration increased trust, coordination and agency	Service collaboration was an approachable goal to begin
	The experience of collaboration built empowerment within the group
	Consistency between institutions improved services
	Personnel changes created challenges for collaboration
The activities of the initiative consistently reflected the community vision	Services became more accessible as the vision developed
	The pop-up hubs go beyond traditional community centres
	Existing community networks were used to reach into community
	Willingness for mental health training helped to build awareness and capacity
OHC coincided with a changed community narrative of hope and agency	The community transitioned from fear to hope
	The community became more willing to solve problems in the context of their strengths
	The community developed the perception that their concerns were heard and addressed
	Positive stories across the community were important to build hope
	Community hope could be threatened in the event of a suicide

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
