# Peer review of "Our Healthy Clarence: A Community-Driven Wellbeing Initiative"

_ijerph, 2019, doi:10.3390/ijerph16193691_

Round 1
Reviewer 1 Report
This is an important paper, with Australia showing more interest than ever in suicide prevention. However, as with many such papers, the result is more about demonstrating a process rather than clear proof of impact. This is not surprising, given time delays in relation to confirmatory data, ethical and other issues. Nevertheless, and to the extent possible, it would be great to see any follow up data on suicides and attempts in the Grafton region provided as part of the article.
At line 195 the authors state that the preceding paragraph raises the issue of community readiness. My feeling is that at least as important here is the issue of sustainability of this kind of community-based approach. In several places in the article this is clearly a factor, where the high motivation of local people to respond to the crisis is affected or worn by time or other factors. What are the implications for this kind of 'bottom-up' approach, in Clarence and more broadly?
The paper should make clearer the limitations which exist around the counting of suicide attempts in Australia, while at the same time providing some sense of their number.
The paper could be improved also be referring to a recent editorial from Science which supported this kind of universal and community approach, here: https://science.sciencemag.org/content/365/6455/725
The paper could also refer to Meadows in relation socio-economic disadvantage: https://www.mja.com.au/journal/2015/202/4/better-access-mental-health-care-and-failure-medicare-principle-universality
The paper could also be more useful if it discussed how the Clarence approach/placed-based method compared and contrasted with the Lifespan approach being undertaken elsewhere. I realise this is not straightforward, but the 'best' thing to do in suicide prevention is currently quite unclear to regions. This paper could enhance its contribution to this discussion by considering its approach in the context of this broader debate.
Reviewer 2 Report
This is an intriguing piece of work that explores an area of community health that could be potentially of interest to many communities that are affected by suicide. However, there are a number of broad problems with the paper that mean I cannot currently support its publication. I list some specific items below.
However, my broad concerns are that I am not convinced that the methodology is robust enough to make the claims that are being made. It is admittedly hard to judge at times as there is a clear lack of clarity throughout the paper as to the evidence that supports the points being made (see specific items below). I got the sense that at time the authors were just using quotes to support their viewpoints or thoughts on these issues rather that engaging with the qualitative material directly.
I did wonder if the language were made clearer throughout if this would improve the paper. For example, it is claimed in line 30 that there “Indicators of impact included increased community optimism” however there are no quantifiable indicators that measured this feature before and after the intervention. The authors thus need to alter their language to make it clear that these are qualitative rather than quantitative changes. The authors list a series of factors that contributed to its successful implementation, but I am still unclear as to how success was measured.
This also links to another problem in that it is claimed this is paper makes use of content analysis of 65 pieces of documentation but I saw no evidence of this being used.
I do hope the authors are able to revise this paper as they have clearly undertaken an interesting project and one which should be robustly evaluated and reported on.
Listed below are some specific issues I picked up:
1) One of the major flaws of the paper is the multiple ways in which the term community is used. It appears to be used interchangeably to refer to the community, as in the people who live there; also the community of workers, often within the same sentence or close by. See lines 100/101, 116/117, 120/121, 365/366, 443/444 for examples though there are many others. This leaves the reader confused as to what ‘community’ is being referred to in a particular sentence. 2) The transparency of the data is poor. Codes are not used to identify and distinguish the participants in the quotes so we are unable to ascertain if all the quotes are in fact from the same person or what their role was in this project. 3) There are repeated problems of rationale in relation to the quotes and the inferences drawn from them which do not appear to match. For example quote on lines 189-193 seem to indicate to me that presence of experts meant there was a withdrawal of the community. But the authors have stated that it raised issues of “community readiness” I don’t see how this follows.Similarly, the quote on lines 253-256 also don’t appear to match the commentary that comes before.
4) The introduction should be revised. There is no mention at all of the cluster/contagion effect associated with suicide that is so relevant to this area of research where a community experiences and unexpected increase. I recognise that they are taking a strengths based approach, however, to not even mention this as a factor in suicide at the community level is remiss. Lines 91 makes references to self-harm rates but does not makes clear that the relationship between suicide and self harm is contested and not straightforward. This tricky relationship is again not referred to in the objectives (line 108/114) Lines 92-94 make repeated use of the term ‘significant’ I am assuming this is not in a statistical sense? If not please reconsider this term 5) Line 96, reference is made to ‘services’ but no indication is made as to how these were identified or what they were. 6) Objectives 5 and 2 seem very similar. What was the rationale for including them both? 7) The limitation of using a purposive sample and of being interviewed by members of the OHC itself seem to provide significant limitations in terms of potential bias that the authors have not addressed. 8) Line 226 contains an error “which are or” 9) Line 261 how do the authors know that the philosophy has remained the same? Furthermore in Line 317 the author describes a decrease in impetus but the quote to me describes much more than this as it is stated that people are not largely “unaware of what OHC is about” 10) Line 323 makes reference to external agencies, but it is not clear who they are. 11) Line 356 and 384 contains data but no source is given to the statistics or how they were gathered. 12) Again, the quote on lines 368-370 does not match or support the commentary on either side. 13) For example, lines 382-388 contain multiple statements that would be better supported by data from the community who took the training. To ask a trainer or sponsor of training whether they think it worked, is poor science and likely to receive the response that is reported, that yes of course it was impactful. As the reader we would require more information in order to be able to assess this statement, 14) In lines 414-418 references are made to the “tone of the newspaper articles” however nothing is given to back this up. Do you have statistical support looking at numbers and content analysis or even qualitative examples of before and after? 15) Line 446 it is stated that has been valuable for “the community see that there is a body…” but how do we know? The community (as I assume it is being used here to mean the body of people who live there) have never been asked.
Reviewer 3 Report
Thank you for the opportunity to review this interesting paper. I have a few comments and suggestions to make:
I am sympathetic to the constraints of wordcount on an academic paper, but I do feel there is a lack of detail in some areas. In particular, the paper refers to, but does not clearly articulate, the community's agency. For example, it isn't clear to me whether the plan was presented to the community as something to comment on, or whether it was developed from the ground up.
Lines 61-62: you state: "A comprehensive approach which encompasses the promotion of mental health and wellbeing, primary and secondary prevention, clinical intervention and postvention will help those who are suicidal, those at risk and may prevent others from becoming so [16,17]." The supporting citations suggest this position is theoretical. Was this position influenced by any primary evidence or local insights? How was it presented to the community as a basis for OHC? Again, I am trying to grasp your understanding of a community process.
Were there key individuals who started this project? Who organised the community meetings? What about conflicts? Are there any data explaining why community attendance at meetings dropped off?
Can you clarify who the stakeholders were, who was on the steering committee and how they all contributed to the process? There seem to have been many people involved but it is not clear from the paper who they were.
Were the community activists representative of the whole community in terms of age/gender/socio-economic status/ethnicity/values? How close knit is the community, and to what extent does OHC reflect the community's shared ambitions?
What is the narrative about funding? Figure 3 illustrates two injections of funding (federal and government): what difference did OHC make to the availability of funds? Was the evaluation a condition of the funding?
In terms of evaluation of impact, can you report any additional discussion about markers of impact? For example, was there any interest in population-level mental health screening?
Were there any theoretical models of health promotion that were considered during the process? If so, did they help? Or was it pretty much atheoretical?
You have not presented any problems or obstacles. Is this because there weren't any?
Finally, a small point: please could you reference the quotations. I am sure your readers will be very interested in how you conducted this study and would want to know who you are quoting.
I realise this might be a lot to ask in a single paper!
Round 2
Reviewer 2 Report
I would like to commend the authors for taking on my comments. This version is much improved and I am happy to approve for publication.